# Social, Demographic, and Psychological Factors Associated with Middle-Aged Mother’s Vocabulary: Findings from the Millennium Cohort Study

**DOI:** 10.3390/jintelligence12060057

**Published:** 2024-05-31

**Authors:** Helen Cheng, Adrian Furnham

**Affiliations:** 1Independent Scholar, London, UK; helenl.cheng@outlook.com; 2Department of Leadership and Organisational Behaviour, Norwegian Business School, Nydalsveien 37, 0484 Oslo, Norway

**Keywords:** maternal vocabulary, maternal education, maternal malaise, parent–child relationship quality, children’s behavioral adjustment, maternal personality traits

## Abstract

Based on a sample of 8271 mothers, this study explored a set of psychological and sociodemographic factors associated with their vocabulary, drawing on data from a large, nationally representative sample of children born in 2000. The dependent variable was maternal vocabulary assessed when cohort members were at fourteen years of age, and the mothers were in their mid-forties. Data were also collected when cohort members were at birth, 9 months old, and at ages 3, 7, 11 and 14 years. Correlational analysis showed that family income at birth, parent–child relationship quality at age 3, maternal educational qualifications at age 11, and maternal personality trait Openness at age 14 were significantly and positively associated with maternal vocabulary. It also showed maternal malaise at 9 months and children’s behavioral adjustment at age 7, and maternal traits Neuroticism and Agreeableness at age 14 were significantly and negatively associated with maternal vocabulary. Maternal age was also significantly and positively associated with vocabulary. Regression analysis showed that maternal age, malaise, parent–child relationship quality, children’s behavioral adjustment, maternal educational qualifications, and traits Openness and Agreeableness were significant predictors of maternal vocabulary, accounting for 33% of total variance. The implications and limitations are discussed.

## 1. Introduction

There are three particular features of this study, which used middle-aged mothers’ vocabulary score as the dependent variable. The first is the question: “what is the relationship between personality and intelligence”, which has been explored using different populations and tests of both factors. We attempted to test two theories (compensation vs. investment) using these data. The second is the relationship between social class, age, and crystallized intelligence as measured by the vocabulary test. The research suggests that there should be a clear correlation between parental social class and intelligence as measured by vocabulary such that higher social class participants would have higher scores ([20]). There would also be a positive age effect as vocabulary, unlike fluid intelligence, increases with age, even in late middle age. Third, this study employed various measures of the parent–child relationship between seven and ten years before the test was administered. The difficulties that arise from these factors influencing parental malaise would influence the mother’s lifestyle and happiness and would be less conducive to activities like reading and social interaction, which would improve vocabulary.

In this study, we chose a number of variables from this publicly available dataset to examine the extent to which variance in IQ, as measured by the proxy variable of vocabulary, could be explained by various psychological and sociodemographic factors. Inevitably, there were other variables that we would like to have examined, but which were not available in the dataset.

## 2. Personality and Intelligence

There is an extensive literature on the relationship between personality and intelligence ([2]; [26]; [48]). Researchers have developed theories as to *why* certain traits (e.g., Conscientiousness, Openness) should be, and are, related to intelligence ([5]; [39]). They are essentially developmental theories and suggest that in adulthood, different traits (Conscientiousness, Openness) would be most strongly related to cognitive ability scores, particularly crystallized intelligence.

One of the most-explored ideas is *Compensation theory*, which argues that Conscientiousness acts as a “coping/reimbursing strategy” for less intelligent but ambitious and competitive people. It is a way of coping in a competitive environment; those who are less intelligent have to work harder than those who are brighter to achieve the same results. Thus, relatively less intelligent individuals may become more methodical, organized, thorough, and persistent (i.e., Conscientious) to compensate for their relative lack of intelligence, particularly in a highly competitive educational or work environment; that is, they can achieve as much as bright people by simply working harder. Alternatively, relatively more intelligent people may tend to get by on their cognitive efficiency rather than strenuous effort or persistent organization. Therefore, IQ and Conscientiousness are negatively correlated. Theoretically this correlation should be less negative for measures of crystallized rather than fluid intelligence given the importance of effort in achieving the former. Various studies have investigated this hypothesis ([34]; [35]; [47]).

Others have argued that Openness is, and should be, most closely related to intelligence. [11] ([11]) hypothesize and found that only aspects from the Openness-to-Experience domain should be empirically associated with intelligence, with the Intellect facet being more strongly associated with intelligence than its counterpart aspect, Openness. This study was replicated by [2] ([2]), who found that the Intellect facet was independently associated with g, verbal, and nonverbal intelligence, while its domain Openness was independently related to verbal intelligence only.

[44] ([44]) proposed an investment theory of adult intelligence, which posits that individual differences in knowledge attainment results from people’s differences in cognitive ability *and* their propensity to apply and invest that ability. These she refers to as investment personality traits. [45] ([45]) identified 34 trait constructs and corresponding scales that refer to intellectual investment which were classified into different trait categories.

It should be noted that there are also reasons to suspect two other traits related to intelligence scores ([7], [8]). It has been suggested that those with higher Neuroticism scores perform less well in all assessments due to test anxiety, which results in a negative correlation. Similarly, extroverts perform less well than introverts in high school and university and thus at a number of IQ tests ([14]).

## 3. Age, Class, Education and Intelligence

There is a large literature which suggests that social class, measured by parental jobs and income, is related to factors including education and intelligence such that higher social class individuals tend to be more intelligent and more successful in educational settings ([6]; [29]). Whilst there are a number of different and contested reasons for this, many studies have confirmed this relationship ([9]; [31]). There are also established age effects such that while fluid intelligence peaks around the early twenties and declines after forty, crystallized intelligence continues to grow until late middle age ([4]).

## 4. Depression and Malaise

There is an extensive literature on the relationship between mental health and intelligence, particularly among those who score very high or low on intelligence tests. Researchers have been interested in particular mental illnesses ([36]). One issue concerns whether some illnesses are more associated with test-taking behavior, which effects scores more than actual IQ ([33]). There is a literature on the relationship between happiness/depression and IQ ([1]), though all researchers acknowledge the complexity of the mechanisms which leads to these results. For instance, do less intelligent people have less healthy lifestyles, or do life circumstances lead to poorer mental health which effects education and IQ development. In this study, we have employed various measures of mental health including malaise and Neuroticism.

Previous studies have shown the strong links between socioeconomic disadvantages and health ([30], [31], [32]; [46]) and maternal depression ([3]; [24]). Maternal depression is linked with parenting ([25]; [43]) and children’s behavioral problems ([12]; [21]; [22]; [27]).

### 4.1. This Study

This study explores a set of psychological and sociodemographic factors associated with maternal vocabulary, drawing on data collected from a large nationally representative sample born in 2000 in the UK, the Millennium Cohort Study (MCS). Vocabulary knowledge is considered to be part of verbal IQ, which is closely related to total IQ ([42]). In the current study, we examine family income, maternal mental health, parent–child relationship, children’s behavioral problems, maternal education, and the Big-Five personality traits in relation to maternal cognitive assessment (i.e., vocabulary). In this sense, we test many hypotheses based on previous psychological and sociological studies concerned with personal and demographic correlates of IQ.

### 4.2. Hypotheses

Based on the literature reviewed above five, hypotheses were formulated. (H1) Family income would be significantly and positively associated with maternal vocabulary; (H2) maternal malaise would be significantly and negatively associated with maternal vocabulary; (H3) parent–child relationship quality would be significantly and positively associated with maternal vocabulary; (H4) children’s behavioral problems would be significantly and negatively associated with maternal vocabulary; (H5) maternal education would be significantly and positively associated with maternal vocabulary; (H6) maternal traits Openness (positively), Extroversion, Conscientiousness, and Neuroticism (negatively) would be significantly associated with maternal vocabulary; (H7) Family income, maternal malaise, parent–child relationship quality, children’s behavioral problems, maternal education, and maternal traits openness and emotional stability would all be significant predictors of the outcome variable.

## 5. Method

### 5.1. Sample

The study draws on data collected for the Millennium Cohort Study (MCS, https://www.data-archive.ac.uk/), a survey of 18,818 babies born between September 2000 and January 2002 into 18,552 families living in the UK ([10]). The first sweep of the Millennium Cohort Study was carried out during 2001 and 2002, when most babies were 9-months old. Data were collected from the parents of the babies via personal interviews and self-completion questionnaires. The sample design allowed for disproportionate representation of families living in areas of child poverty. Due to disproportionate sampling, special weights have to be applied in analyzing the data ([38]). The families were followed up. The following analyses are based on data from surveyr at 9 months of age and 3, 7, 11, and 14 years. Data were collected from parents (mainly mothers) via personal interview sand self-completion questionnaires. In 2015, when cohort members were at 14 years of age, 11,872 cohort members from 11,726 families took part in the survey (response = 76%). A total of 10,282 mothers completed a vocabulary assessment. Informed consent was obtained from all subjects involved in the study ([41]). The following analyses are based on 8271 mothers and their children, for whom we have complete data. In comparison to the original sample, the analytic sample contains relatively more socially privileged and better-educated mothers, and slightly more boys.

### 5.2. Measures

*Maternal age*: Information on maternal age were provided by mothers through interviews in several sweeps. The maternal age when cohort members were born was used in the study. The mean age was 29.65 (SD = 5.38), ranging from 14 to 47 years old. In effect, they were around 43 years old when the study was run.*Family income*: Family income of the household was reported at birth. The logged family income is used for the analyses.*Maternal malaise*: This was assessed when cohort members were 9 months old. A shortened 9-item version of the Rutter Malaise Inventory ([40]) was used, which is a self-completion instrument measuring depression, anxiety, and psychosomatic illness ([40]), and it correlates significantly with previously diagnosed and currently treated depression. It has been shown to be relatively stable over time ([15]). Cronbach’s alpha was 0.75.*Parent–child relationship*: This was assessed when cohort members were at 3 years of age using the Pianta scale ([37]), comprising 15 items on a 5-point Likert scale (1 = definitely not apply; 5 = definitely applies). Example items: “I share an affectionate, warm relationship with my child”; “dealing with my child drains my energy”. Information was collected at age 3 using the mother’s report. Responses were summed, with a high score indicating a better parent–child relationship. Cronbach’s alpha was 0.77.*Children’s behavioral problems*: This was measured with the Strength and Difficulties Questionnaire (SDQ) via the mother’s report. Mothers were interviewed when cohort members were at 7 years of age. Behavioral adjustment at age 7 is a behavioral screening questionnaire for 3-to-16 year olds ([17]; [18], [19]). It consists of 25 items, assessed via parental report, generating scores for five subscales measuring hyperactivity, emotional symptoms, conduct problems, peer problems, and prosocial behavior. The total difficulties score does not incorporate the prosocial scale, which measures prosocial behavior ([18]). Thus, the four behavioral problems subscales were used as the outcome measures. Each SDQ item has three possible answers, which are assigned a value: 0 = not true; 1 = somewhat true; or 2 = certainly true. The 20-item SDQ total score was used in the following analysis. Cronbach’s alpha for hyperactivity was 0.79; for emotional symptoms, it was 0.66; for conduct problems, it was 0.62; and for peer problems, it was 0.61. Cronbach’s alpha for the SDQ total score was 0.73.*Maternal education:* This was measured when cohort members were at age 11. Mothers were asked about their highest academic or vocational qualifications. Responses are coded according to the six-point scale of National Vocational Qualifications levels (NVQ), which ranges from “none” to “university degree/higher”/equivalent NVQ 5 or 6.*Maternal personality traits*: The Big-Five personality traits of Extroversion, Neuroticism/Emotional Stability, Agreeableness, Conscientiousness, and Openness were assessed when cohort members were at age 14 years. We used a shortened version (3 items for each trait) of the International Personality Item Pool (IPIP) ([16]), a self-completion questionnaire. Responses (7-point scale: 1 is “does not apply to me at all” and 7 is “applies to me perfectly”). Cronbach’s alpha was 0.60 for Extroversion, 0.67 for Neuroticism/Emotional Stability, 0.53 for Agreeableness, 0.52 for Conscientiousness, and 0.66 for Openness.*Maternal vocabulary*: This was assessed when cohort members were at 14 years of age. It is a word activity assessment measuring knowledge of vocabulary. Mothers were asked to complete the word activity during the interviewer visit. It measures respondents’ understanding of the meaning of words. The assessment involved presenting the respondent with a list of target words, each of which had five other words next to them. The respondent had to select, from the five options, the word which meant the same, or nearly the same, as the target word (i.e., synonyms). Each respondent had a list of 20 target words. The assessment was carried out on the interviewer’s tablet.

## 6. Results

### 6.1. Correlational Analysis

Table 1 shows the correlations and the means and SDs of all variables in this study. Maternal vocabulary was significantly and positively associated with family income, parent–child relationship quality, maternal education, and maternal trait Openness, and significantly and negatively associated with maternal malaise, children’s behavioral problems, and trait Neuroticism (*p* < .05 to *p* < .001). Thus, (H1) to (H6) were supported. Maternal age at birth was also significantly and positively associated with maternal vocabulary. Among the Big-Five personality traits, Extroversion and Conscientiousness were not related to maternal vocabulary (omitted in further analysis). The strongest associations were between maternal education and maternal vocabulary (*r* = 0.46; *p* < .001), followed by maternal age and family income (*r* = 0.39 and *r* = 0.37, respectively; *p* < .001).

### 6.2. Regressional Analysis

Table 2 shows the results of the regressional analysis. In Model 1, all six psychological variables—maternal malaise, parent–child relationship quality, children’s behavioral problems, and traits Neuroticism, Agreeableness, and Openness—were significant predictors of maternal vocabulary (Beta ranged from −0.03 to 0.17; *p* < .05 to *p* < .001), accounting for 7% of the variance. In Model 2, after entering the three sociodemographic variables into the equation, trait Neuroticism ceased to be a significant predictor of the outcome variable, whilst maternal age, family income, and maternal education (Beta ranged from 0.13 to 0.35; *p* < .001) accounted for 26% of variance, thus explaining 33% of the total variance.

## 7. Discussion

This study confirmed and extended the previous findings in the area. Among the psychological factors, the strongest predictor of maternal vocabulary was trait Openness (Beta = 0.10; *p* < .001). This is in line with previous findings which have shown the significant and positive association between Openness and numerous measures of intelligence ([13]). Indeed, it supports the investment hypothesis, which would suggest that a measure of vocabulary would be particularly affected by an investment into learning through consumption of various media and social interactions. Open people are curious and interested in new ideas and new knowledge, which inevitably leads to greater knowledge (crystallized intelligence).

The associations between Conscientiousness and cognitive ability and between Agreeableness and cognitive ability are not unequivocal in previous studies. Correlation coefficients are usually very modest. In the current study, Conscientiousness was not associated with maternal vocabulary (*r* = 0.01, ns), though previous findings have shown this trait to be significantly and positively association with occupational attainment, which is largely related to cognitive ability ([13]). One of the explanations of the association between Conscientiousness and cognitive ability might be due to some facets of Conscientiousness ([28]) that affect vocabulary assessment, which have a time restriction. This, therefore, does not provide support for the *Compensation theory.*

In the current study, Agreeableness was significantly and negatively associated with maternal vocabulary (Beta = −0.06; *p* < .001). It is not clear why agreeable mothers tend to have lower scores on vocabulary assessment, though it could be related to a preference for particular reading material or social interaction patterns.

The current study also showed that maternal malaise and parent–child relationship quality were modestly but significantly associated with maternal vocabulary. Intelligent mothers tend to be less prone to mental health problems and are more likely to have good relationships with their children. It might be that intelligent mothers tend to have an authoritative (rather than authoritarian or permissive) rearing style in their children’s upbringing and to nurture children’s behavioral adjustment.

There was a significant and positive association between maternal age when cohort members were born and maternal vocabulary, suggesting that intelligent mothers tend to have children relatively later in their lives. Indeed, we know that vocabulary is related to age; therefore, this may simply be evidence of an age effect.

Mothers’ age was significantly and positively associated with family income (*r* = 0.39, *p* < .00), and more mature mothers are more able to provide their children with better living standards and learning facilities compared with younger mothers, especially teenager mothers. There are data to suggest that social class is related to the age of marriage, with lower SES people being associated with earlier marriages ([23]).

Educational achievement was the strongest predictor of maternal vocabulary (Beta = 0.35; *p* < .001). The association might be bi-directional: intelligent children are more likely to obtain higher educational qualifications, and educational qualifications lead to higher occupational attainment in a more mentally stimulating working environment, which would help maintain individuals’ cognitive abilities.

### Limitations

Whilst it is very desirable to be able to explore a dataset such as this, it is also not always clear as to the causal relationships between the variables. Thus, our major variable of interest, namely, vocabulary (acting as a proxy for IQ), could be understood as either or both an outcome or a predictor variable. Thus, trait Openness could be seen to lead to a higher vocabulary because of curiosity; but equally, a greater vocabulary could increase Openness because of wider access to new and stimulating material. Similarly, in our regressions, we used vocabulary as our independent (criterion) variable, and factors such as malaise as dependent (predictor) variables, but we could have reversed this by looking at the effect of vocabulary and other factors on malaise.

As with all research using cohort studies, the variables used in the study are constrained by the availability of the data. Many of the measures—for instance, those of personality—were very short, which could threaten their validity. Another limitation is the attrition of respondents over time. Since sample attrition is greatest amongst individuals in more deprived circumstances, our results may thus be a conservative estimate. Another limitation is that all measurements were made via mother-report, with the concomitant problems of method invariance.

Further, it would be desirable to have maternal cognitive ability assessed earlier, i.e., at the birth of cohort members. To build a robust causal model of the participants’ IQ (vocabulary) in their mid-forties, it would have been desirable to have more data on such things as the number of children they had, family dynamics, and partner support, as well as occupational status history and other personality and motivational variables like ambition and tolerance for ambiguity ([7], [8]).

## Figures and Tables

**Table 1 jintelligence-12-00057-t001:** Pearson product–moment correlations of maternal vocabulary and other variables in the study.

	Variables	Mean (SD)	1	2	3	4	5	6	7	8	9	10	11	12
1.	Maternal vocabulary at age 14	12.20 (3.96)	_											
2.	Maternal age at birth	29.65 (5.38)	**0.39 *****	_										
3.	Logged family income at birth	2.51 (0.30)	**0.37 *****	0.39 ***	_									
4.	Maternal malaise at 9 months	2.51 (0.30)	**−0.10 *****	−0.11 ***	0.16 ***	_								
5.	Parent–child relationship quality at age 3	65.01 (6.45)	**0.06 *****	0.11 ***	0.11 ***	−0.27 ***	_							
6.	Children’s behavioral problems at age 7	6.59 (4.99)	**−0.17 *****	−0.19 ***	−0.23 ***	0.26 ***	0.39 ***	_						
7.	Maternal education at age 11	3.09 (1.21)	**0.46 ****	0.22 ***	0.36 ***	−0.06 ***	0.06 ***	−0.18 ***	_					
8.	Maternal Extroversion at age 14	14.42 (3.90)	**−0.01**	0.02	0.08 ***	−0.10 ***	0.08 ***	−0.11 ***	0.03 *	_				
9.	Maternal Neuroticism at age 14	11.85 (4.11)	**−0.04 ****	−0.09 ***	−0.10 ***	0.32 ***	0.19 ***	0.21 ***	−0.07 ***	0.25 ***	_			
10.	Maternal Agreeableness at age 14	18.10 (2.63)	**−0.06 *****	0.01	−0.02	−0.05 ***	0.12 ***	−0.08 ***	−0.01	0.16 ***	−0.07 ***	_		
11.	Maternal Conscientiousness at age 14	17.66 (2.99)	**0.01**	0.04 **	0.09 ***	−0.13 ***	0.14 ***	−0.15 ***	0.05 ***	0.26 ***	−0.15 ***	0.38 ***	_	
12.	Maternal Openness at age 14	13.93 (3.73)	**0.14 *****	0.09 ***	0.09 ***	−0.03 *	0.07 ***	−0.09 ***	0.16 ***	0.29 ***	−0.10 ***	0.19 ***	0.24 ***	_

Note: Variables were scored such that a higher score indicated the mother’s higher age when giving birth to cohort members, a higher score on family income, higher scores on maternal malaise, higher scores on positive parent–child relationship, higher scores on children behavioral problems, higher scores on traits maternal Extroversion, Neuroticism, Agreeableness, Conscientiousness, and Openness, and higher scores on maternal vocabulary. Bold coefficients indicate the associations between the outcome variables and other variables examined in the study. Analysis is weighted using UK sampling weight. * *p* < .05; ** *p* < .01; *** *p* < .001.

**Table 2 jintelligence-12-00057-t002:** Predicting maternal vocabulary.

Measures	Model 1		Model 2		
	Beta	*t*	Beta	*t*	*p* †
*Psychological factors*					
Maternal malaise	−0.05	3.09 **	−0.04	2.79 **	.005
Parent–child relationship quality	0.04	2.63 **	0.03	2.72 **	.007
Children’s behavioral problems	−0.16	11.26 ***	−0.06	4.33 ***	<.001
Neuroticism	−0.03	2.07 *	0.02	1.67	.056
Agreeableness	−0.09	6.54 ***	−0.06	5.43 ***	<.001
Openness	0.17	12.32 ***	0.10	7.40 ***	<.001
*Sociodemographic factors*					
Maternal age			0.25	20.77 ***	<.001
Logged family income			0.13	10.08 ***	<.001
Maternal education			0.35	28.95 ***	<.001
*Variance explained*	*R*^2^ *adjusted* = 0.07		*R*^2^ *adjusted* = 0.33		
	*F* = 49.02 ***		*F* = 242.07 ***		

Note. * *p* < .05; ** *p* < .01; *** *p* < .001. † Significance levels in the final model. Analysis is weighted using UK sampling weight.

## Data Availability

Further details of the data available from the main survey are available from the CLS website and, in particular, the MCS Guide to the Datasets at www.cls.ioe.ac.uk/mcs, accessed on 5 April 2024.

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
