# Peer review of "Social, Demographic, and Psychological Factors Associated with Middle-Aged Mother’s Vocabulary: Findings from the Millennium Cohort Study"

_jintelligence, 2024, doi:10.3390/jintelligence12060057_

Round 1

Reviewer 1 Report

Comments and Suggestions for Authors

Abstract: On initial reading, it is unclear whether “participant” refers to the 8,271 mothers in the sample or their offspring. How many offspring are in the sample? 

Introduction:

Lines 24-36: The first paragraph of the introduction is not clearly written. “is the relationship between personality and intelligence” Do you mean “What is the relationship between personality and intelligence”? I have trouble parsing what the authors are studying and what research questions are being asked.

It is unclear to me how “Compensation theory” can be thought of as a longitudinal theory.

Lines 56-57: “Theoretically…” Is there research to support this statement about crystalized intelligence? 

Line 81: Incomplete (?) citations

Lines 77-83: Lots of statements in this paragraph that need sources. Without references they appear to be the authors’ opinions. E.g., fluid intelligence peaking in early twenties but crystallized increasing until late middle age.

Lines 87-88: Maternal depression is linked with parenting how? What is “parenting” in this context?

4.1 This study: None of the previous part of the introduction discusses vocabulary specifically as a predictive or predicted variable in the context of personality, intelligence, malaise, etc. It is difficult to see how the previously cited literature leads up to the set of current hypotheses proposed by the authors.

Comments on the Quality of English Language

Flow and coherency must be improved--I had trouble understanding what the authors are doing in many cases.

Author Response

Abstract: On initial reading, it is unclear whether “participant” refers to the 8,271 mothers in the sample or their offspring. How many offspring are in the sample? 

Clarified as requested

Introduction:

Lines 24-36: The first paragraph of the introduction is not clearly written. “is the relationship between personality and intelligence” Do you mean “What is the relationship between personality and intelligence”? I have trouble parsing what the authors are studying and what research questions are being asked.

Ok…..revised, and hopefully much clearer

It is unclear to me how “Compensation theory” can be thought of as a longitudinal theory.

Have attempted to explain this

Lines 56-57: “Theoretically…” Is there research to support this statement about crystalized intelligence? 

Clarified this

Line 81: Incomplete (?) citations

Corrected

Lines 77-83: Lots of statements in this paragraph that need sources. Without references they appear to be the authors’ opinions. E.g., fluid intelligence peaking in early twenties but crystallized increasing until late middle age.

Where appropriate I have added references

Lines 87-88: Maternal depression is linked with parenting how? What is “parenting” in this context?

Attempted to explain this

4.1 This study: None of the previous part of the introduction discusses vocabulary specifically as a predictive or predicted variable in the context of personality, intelligence, malaise, etc. It is difficult to see how the previously cited literature leads up to the set of current hypotheses proposed by the authors.

A good point which we have attempted to clarify

Reviewer 2 Report

Comments and Suggestions for Authors

I found this to be an interesting and compelling manuscript. At the same time, I think there are several aspects that can and should be improved. Specifically, the rationale for choosing these specific predictor variables needs to be made clearer and more interpretation of the results needs to be provided. As it stands, the authors report some interesting correlational results, but how we as readers should be make sense of these results, especially with regard to causality, is less clear. Ideally, the authors should provide a unified approach, streamlining the hypotheses into a whole that makes sense from a theoretical perspective, that needs to be focused on also in the discussion. In the following, I provide some additional comments specific to certain parts of the text.

Introduction
"We attempted to test two theories using this data." Maybe hint at what these theories might be about, or leave this point for later. The theories should in general be more prominent across the manuscript, I think.

"The data suggest that there would be a clear correlation between parental social class and intelligence as measured by vocabulary such that higher social class participants would have higher scores"
I think the authors allude to previous studies here? This should be made clear.

line 25 typo "Conscientious"
The authors should provide references regarding "compensation theory" and related empirical findings.

"Theoretically this correlation should be higher for measures of crystalised rather than fluid intelligence given the importance of effort in achieving the former"
I had a hard time understanding this wording. I think the authors mean "less negative", and not "a stronger negative correlation" here?

line 81: (Deary et al……; Marmot….) references are incomplete and missing for the age effects that are described next.

"4. Depression and Malais"
No links to cognitive abilities are described here. Thus it is hard to use this section to argue for the hypotheses that follow (H2, H3, H4, H7). It would be very important to make this section much stronger, as these questions are at the core of the presented research.

"The following analyses are based on 8,271 mothers and their children for whom we have complete data"
Why did the authors decide to only include complete cases? I think given the methods that are available nowadays for dealing with missing data, this might be a waste of potentially useful data (roughly 20% of those who completed the main outcome measure). At least the authors should show that the results are similar if all data are used.

The Big Five measure used is very short, this at the very least needs to be discussed.

Table 1: The correlation between vocabulary and malaise is positive here, other than what was described in the text?

Discussion:
"In the current study, Agreeableness is significantly and negatively associated with aternal vocabulary (Beta = -.06, p<.001). It is not clear why agreeable mothers tend to have higher scores on vocabulary assessment though it could be related to a preference for particular reading material, or better social interaction."
As the association is negative, lower (not higher) agreeableness scores should go along with high vocabulary scores.

"The significant and positive association between maternal age when cohort members were born and maternal vocabulary suggesting that intelligent mothers tend to have children relatively later in their lives. "
As vocabulary is related to age (irrespective of whether women become a mother or not), might this not simply be an age effect?

"There is data to suggest that social class is related to age of marriage with lower SES people being associated with earlier marriages. " Reference missing.

"Further, it would be desirable to have maternal cognitive ability assessed earlier, at the birth of cohort members."
Indeed. As it stands, it is very hard to interpret some of the results, as we don't know if the vocabulary (or what it stands for) should be understood as an outcome or a predictor in most cases. In the discussion, maternal malaise and parent-child relationship are presented more as outcomes of intelligence. This makes sense, but it also makes the use of the regression model (with the assumed causality reversed) slightly questionable. Overall, I think the discussion part needs to be greatly expanded, including a true discussion of what the limitations might mean for the interpretation of the study and individual results. Specific emphasis should be placed on the assumed causal relationships between the modeled parameters, with references to the literature provided.

Comments on the Quality of English Language

Please check again for minor mistakes (spelling, grammar).

Author Response

Rev2

I found this to be an interesting and compelling manuscript. At the same time, I think there are several aspects that can and should be improved. Specifically, the rationale for choosing these specific predictor variables needs to be made clearer and more interpretation of the results needs to be provided. As it stands, the authors report some interesting correlational results, but how we as readers should be make sense of these results, especially with regard to causality, is less clear. Ideally, the authors should provide a unified approach, streamlining the hypotheses into a whole that makes sense from a theoretical perspective, that needs to be focused on also in the discussion.

A helpful overview which we have attempted to correct

In the following, I provide some additional comments specific to certain parts of the text.

Introduction
"We attempted to test two theories using this data." Maybe hint at what these theories might be about, or leave this point for later. The theories should in general be more prominent across the manuscript, I think.

Done as suggested

"The data suggest that there would be a clear correlation between parental social class and intelligence as measured by vocabulary such that higher social class participants would have higher scores"
I think the authors allude to previous studies here? This should be made clear.

Done as suggested

line 25 typo "Conscientious"

Corrected
The authors should provide references regarding "compensation theory" and related empirical findings.

Done as suggested

"Theoretically this correlation should be higher for measures of crystalised rather than fluid intelligence given the importance of effort in achieving the former"
I had a hard time understanding this wording. I think the authors mean "less negative", and not "a stronger negative correlation" here?

Correct and changes made

line 81: (Deary et al……; Marmot….) references are incomplete and missing for the age effects that are described next.

Corrected

"4. Depression and Malais"
No links to cognitive abilities are described here. Thus it is hard to use this section to argue for the hypotheses that follow (H2, H3, H4, H7). It would be very important to make this section much stronger, as these questions are at the core of the presented research.

Ok..tried to do just this

"The following analyses are based on 8,271 mothers and their children for whom we have complete data"
Why did the authors decide to only include complete cases? I think given the methods that are available nowadays for dealing with missing data, this might be a waste of potentially useful data (roughly 20% of those who completed the main outcome measure). At least the authors should show that the results are similar if all data are used.

We thought this number was quite sufficient to test all hypotheses

The Big Five measure used is very short, this at the very least needs to be discussed.

Agreed….and done as suggested

Table 1: The correlation between vocabulary and malaise is positive here, other than what was described in the text?

Checked and corrected

Discussion:
"In the current study, Agreeableness is significantly and negatively associated with maternal vocabulary (Beta = -.06, p<.001). It is not clear why agreeable mothers tend to have higher scores on vocabulary assessment though it could be related to a preference for particular reading material, or better social interaction."
As the association is negative, lower (not higher) agreeableness scores should go along with high vocabulary scores.

Thank you..this section revisited and improved

"The significant and positive association between maternal age when cohort members were born and maternal vocabulary suggesting that intelligent mothers tend to have children relatively later in their lives. "
As vocabulary is related to age (irrespective of whether women become a mother or not), might this not simply be an age effect?

Yes indeed and point noted

"There is data to suggest that social class is related to age of marriage with lower SES people being associated with earlier marriages. " Reference missing.

Now included

"Further, it would be desirable to have maternal cognitive ability assessed earlier, at the birth of cohort members."
Indeed. As it stands, it is very hard to interpret some of the results, as we don't know if the vocabulary (or what it stands for) should be understood as an outcome or a predictor in most cases. In the discussion, maternal malaise and parent-child relationship are presented more as outcomes of intelligence. This makes sense, but it also makes the use of the regression model (with the assumed causality reversed) slightly questionable. Overall, I think the discussion part needs to be greatly expanded, including a true discussion of what the limitations might mean for the interpretation of the study and individual results. Specific emphasis should be placed on the assumed causal relationships between the modeled parameters, with references to the literature provided

Again, very helpful

Round 2

Reviewer 2 Report

Comments and Suggestions for Authors

I commend the authors for addressing the points I have raised in my review. I feel like the manuscript has profited from the changes made. As a side note, I would have found it very helpful if the authors had directly referred to the changes made within the response letter and provided more context there.

I still think that the manuscript could profit from a more streamlined theoretical approach (the two models mentioned at the beginning are not a central part of the discussion) and more causal reasoning. The authors now acknowledge that the causal relationships are unclear. I would have liked the authors to take a slightly more bold approach, by explicitly stating their theoretical assumptions for building the model as they conceptualized it (why these predictors and not others, why use them as predictors). This way, the paper could gain an additional dimension to the partly explorative reporting of the associations of the vocabulary test.

Author Response

I made four additions in blue to answer the points